# Transcriptome Analysis of *Chenopodium album* in Response to Infection by *Botrytis* Strain HZ-011

**DOI:** 10.3390/microorganisms13092177

**Published:** 2025-09-18

**Authors:** Haixia Zhu, Le Zhang, Yongqiang Ma, Lu Hou

**Affiliations:** 1Academy of Agriculture and Forestry Sciences, Qinghai University, Xi’ning 810016, China; zhuhaixia0101@163.com (H.Z.); 13919513590@163.com (L.Z.); mantou428@163.com (L.H.); 2Scientific Observing and Experimental Station of Crop Pest in Xining in Ministry of Agriculture, Xi’ning 810016, China; 3Key Laboratory of Agricultural Integrated Pest Management in Qinghai Province, Xi’ning 810016, China

**Keywords:** transcriptome sequencing, *Botrytis* strain HZ-011, *Chenopodium album*, key genes

## Abstract

This study conducted a transcriptome sequencing analysis of the interaction between *Chenopodium album* and *Botrytis* strain HZ-011 to identify genes involved in the response to fungal infections and elucidate the molecular mechanisms underlying the interaction. High-throughput RNA-seq technology was employed to analyze the transcriptomes of *C. album* leaves at 1, 4, and 5 days post-inoculation (dpi) with *Botrytis* strain HZ-011. The results revealed 11,645 differentially expressed genes (DEGs) at 1 dpi, including 7399 upregulated and 4246 downregulated genes; 11,285 DEGs at 4 dpi (7801 upregulated and 3484 downregulated); and 9976 DEGs at 5 dpi (7723 upregulated and 2253 downregulated). GO functional analysis indicated that downregulated DEGs were significantly enriched in chloroplast and plastid functional expression at 1, 4, and 5 dpi. Following infection by *Botrytis* strain HZ-011, downregulated genes were significantly enriched in pathways related to photosynthesis, including photosynthetic pathways, light-harvesting antenna proteins, and carotenoid biosynthesis. This suggests that the photosynthetic process in *C. album* was markedly inhibited, disrupting nutrient supply and leading to herbicidal effects. Notably, genes such as *PSB28*, *PSBP*, *CAP10A*, and *CRTL-E-1* were significantly enriched in these pathways, indicating their potential roles in the herbicidal mechanism. These findings provide a foundation for understanding the herbicidal activity of strain HZ-011 and identifying potential targets for developing novel microbial herbicides.

## 1. Introduction

*Chenopodium album* L., is an annual broadleaf weed belonging to the family Amaranthaceae. It is a pervasive and highly detrimental weed worldwide, thriving in subtropical, tropical, and temperate regions due to its robust reproductive capacity and allelopathic effects [1,2,3]. Its uncontrolled growth severely impacts crop quality and yield [4]. In Qinghai Province, *C. album* is a dominant weed that disrupts crop growth, reduces yields, and threatens agricultural ecosystem balance. Current control methods include mechanical weeding and chemical herbicides [5], but extensive herbicide use poses significant environmental risks [6,7]. Consequently, safer and more sustainable biological control strategies are gaining attention, particularly in the context of green pest management.

*Botrytis*, a pathogenic fungus within the Ascomycota phylum, exhibits a broad host range. It secretes toxins, cell wall-degrading enzymes, and cellulases, enabling it to breach plant defenses and cause disease [8,9]. Recent studies highlight its herbicidal potential against certain weeds [10,11]. For instance, Wang et al. [12] demonstrated that toxins extracted from *Botrytis cinerea* induced 100% pathogenicity in seedlings of *Amaranthus retroflexus*, *Ipomoea nil*, and other dicot weeds. Li Chungge [13] isolated 11 *Botrytis* mstrains from various hosts, revealing that their metabolites inhibited the growth of *A. retroflexus*, *Ipomoea hederacea*, and *C. album*, with the strongest suppression observed in *A. retroflexus*. Zhang Jinlin et al. [14] mutagenized *Botrytis* strain BC4 and identified BC4-1, a mutant whose ethyl acetate extracts completely inhibited the seed germination of *Digitaria sanguinalis* and *A. retroflexus*.

Although scholars worldwide have screened numerous biocontrol bacteria with weed-suppressing potential, few studies have comprehensively investigated the molecular mechanisms underlying their weed suppression. Elucidating the molecular mechanisms by which biocontrol bacteria suppress weeds could provide opportunities for identifying potential targets for herbicides [15,16,17]. Transcriptome sequencing technology enables the acquisition of extensive gene information, gene function, gene family identification, and metabolic pathways at the mRNA level, linking significantly expressed genes to specific biological functions [18]. This technology offers unparalleled advantages in analyzing plant–pathogen interactions. Zhang Ke et al. [19] employed transcriptomics to analyze the main biological metabolic processes and screen out the molecular processes of the interaction between *Rhizoctonia solani* and *Zoysia japonica*, based on the transcriptome of *R. solani* infecting *Z. japonica*. The results showed that Ca^2+^ regulation and phosphorylation pathways were closely related to the response of *Z. japonica* to *R. solani*. Liu Xiaofang et al. [20] employed RNA-seq technology to analyze the transcriptome of *C. album* leaves infected by *Aureobasidium pullulans*; results indicated that pathogen infection altered expression levels of key genes in pathways including plant MAPK signaling, plant–pathogen interactions, and plant hormone signaling. This modification enhanced *C. album*’s antioxidant capacity, ultimately achieving weed control.

*Botrytis* strain HZ-011, isolated from naturally infected *Rumex patientia* leaves by our team, exhibits potent herbicidal activity against *C. album* and other weeds [21]. Preserved at the China General Microbiological Culture Collection Center (CGMCC No. 23894), this strain has undergone preliminary studies on isolation, pathogenicity, fermentation optimization, and herbicidal compound identification [22]. However, the molecular mechanisms underlying *C. album*’s response to HZ-011 infection remain unexplored. This study employs transcriptome sequencing to investigate these mechanisms from both pathogen and host perspectives, providing theoretical insights into *Botrytis*–weed interactions.

## 2. Materials and Methods

### 2.1. Materials

Fungal strain: *Botrytis* strain HZ-011, isolated from infected *Rumex patientia* leaves in July 2020, was preserved at the Key Laboratory of Integrated Pest Management, Qinghai Academy of Agricultural and Forestry Sciences.

Weed material: Healthy *C. album* seedlings (4–6 leaves) were collected from experimental fields (101.74° E, 36.56° N), transplanted into pots, and acclimated in a greenhouse at 25 °C for 7 days.

Media Types and Formulations PDA Medium: Potato 200 g/L, glucose 20 g/L, agar powder 20 g/L, neutral pH.

### 2.2. Methods

#### 2.2.1. HZ-011 Spore Suspension Preparation

Inoculate the HZ-011 grape spore strain, stored at 4 °C, onto PDA agar plates and incubate at 25 °C for 7 days to activate. Select PDA plates with successful activation. Add 5 mL of sterile water to each plate, scrape the mycelium and filter using sterile filter paper, centrifuge the suspension at 4000 rpm for 5 min, transfer the supernatant to a new sterile centrifuge tube, and use a hemocytometer to calculate the initial spore suspension concentration under a microscope. Then, dilute the initial spore suspension with sterile water to 1 × 10^8^ CFU/mL for later use.

#### 2.2.2. HZ-011 Spore Suspension Inoculation of Weed Seedlings

Select healthy 4–6-leaf-stage quinoa seedlings with no damage and transplant them into flowerpots (14 cm in diameter) for cultivation in a greenhouse for one week. Inoculate each pot with 25 mL of the HZ-011 strain spore suspension prepared in Section 2.1 using both injection and spray methods. Maintain humidity at 70% relative humidity for 24 h post-inoculation; then, allow to incubate. Collect tissue samples at the disease–healthy interface of diseased leaves at 1 day, 4 days, 5 days post-inoculation; tissue samples were collected from the diseased–healthy transition zone of infected leaves. Leaves inoculated with sterile water on day 0 and HZ-011 strain cultured on PDA plates served as controls. Each sample consisted of 1.5–2.0 g of leaf tissue from the treatment group and control group, with three biological replicates per treatment. Samples were stored at −80 °C.

#### 2.2.3. RNA Extraction, Library Construction, and Sequencing

Total RNA extraction from treatment and control samples was performed using the Trizol™ Reagent Kit (Invitrogen, Waltham, MA, USA). RNA quality was assessed on a bioanalyzer, and total RNA integrity was evaluated by 1% RNase-free agarose gel electrophoresis. Following total RNA extraction, mRNA was enriched using mRNA Capture Beads magnetic beads. The enriched mRNA was then fragmented at elevated temperatures. This fragmented mRNA served as a template for synthesizing the first strand of cDNA within a reverse transcriptase mix. During second-strand cDNA synthesis, end repair and A-tailing were completed. Adapters were subsequently ligated, and target fragments were purified using HieffNGS^®^ DNA Selection Beads. Following PCR library amplification, sequencing was performed by Guangzhou Gedi Biotechnology Co., Ltd. (Guangzhou, China) using the Illumina NovaSeq X Plus platform (San Diego, CA, USA).

#### 2.2.4. Clean Reads Filtering and Data Assembly

To obtain high-quality clean reads, the raw reads were further filtered using fastp (version 0.18.0). The read filtering steps are as follows:Remove reads containing adapters from reverse transcription.Remove reads with an N proportion greater than 10%.Remove reads consisting entirely of A bases.Remove low-quality reads (where bases with a quality score of Q ≤ 20 account for more than 50% of the entire read)

Then, the short text assembly program Trinity software v2.15.1 was used for transcriptomic data reassembly. The assembly results were quality-assessed using N50 values, sequence length, and BUSCO v6.0.0.

#### 2.2.5. Single-Gene Expression Analysis and Basic Annotation

Calculate the expression level of a single gene and normalize it to RPKM (reads per million reads per kb), using the following formula:RPKM = (1,000,000 × C)/(N × L/1000)(1)

C is the number of reads uniquely mapped to single gene A, and N is the total number of reads uniquely mapped to all single genes. L is the length of gene A (number of bases).

Using the NCBI BLAST+ 2.13.0+ with an E-value threshold of 1 × 10^−5^, Unigene sequences were aligned with the NCBI Non-Redundant Protein Nr database (20200514), Swiss-Prot protein database (20200514), KEGG database (release101), and COG/KOG database. Protein functional annotations were obtained based on the best alignment results.

#### 2.2.6. Reference Sequence Alignment

Use the alignment tool Bowite to map high-quality clean reads to the reference sequence and calculate the mapping ratio.

Mapping ratio = (number of uniquely mapped reads + number of multiplied mapped reads)/total number of reads.

#### 2.2.7. Differentially Expressed Genes (DEGs) Screening and Enrichment Analysis

The differential expression analysis of RNA between different treatments was performed using the edgeR software 3.40.x. Genes with an FDR parameter below 0.05 and an absolute fold change of ≥2 were considered differentially expressed genes. The BLASTx program, GSEA software 4.3.x, and MSigDB software 4.3.x were used to make comparisons with the GO database and KEGG database to perform gene ontology (GO) enrichment, metabolic pathway enrichment, and genome enrichment analysis (GESA).

####  2.2.8. qRT-PCR Validation of Differentially Expressed Genes in the Transcriptome

The qRT-PCR samples and transcriptomic sequencing samples were from the same batch. RNA reverse transcription and cDNA synthesis were performed using the FastKing gDNA Dispersing RT SuperMix kit from TianGen Bio-Chem Technology Co., Ltd. (Beijing, China). Twenty significantly altered differentially expressed genes were selected from the control transcriptomic dataset to validate the transcriptional changes following infection of *Chenopodium* leaves by the Grapevine HZ-011 strain. Primer design was performed using Primer 6.0 software based on the CDS sequences of the differentially expressed genes, and primers were synthesized by Shanghai Sangon Biotechnology Co., Ltd. (Shanghai, China). Quantitative analysis of differentially expressed genes was conducted using real-time quantitative PCR and the 2^−ΔΔCt^ method. The housekeeping gene Actin was selected as the internal control gene, and fluorescent quantitative validation experiments were completed using a fluorescent quantitative pre-mixed reagent kit (SYBR Green) (Beijing Tiangen Biochemical Technology Co., Ltd., Beijing, China). qRT-PCR was performed in a 25 μL system with the following reaction program: 95 °C pre-denaturation for 2 min, 1×; 95 °C denaturation for 5 s, 60 °C annealing/extension for 32 s, 40×. The qRT-PCR reaction system and transcriptomic qRT-PCR primer sequences are shown in Table 1 and Table 2. Set up template-free controls; each sample’s qRT-qPCR analysis includes three technical replicates.

## 3. Results

### 3.1. Data Quality Control

To ensure data reliability, raw sequencing reads were filtered to remove low-quality data prior to analysis. As shown in Table 3, the average number of raw reads across 12 samples was 43,605,502. After filtering, an average of 43,426,401 high-quality reads (6,425,603,137 bp) were retained. Q20 and Q30 values represent base call error rates of 1% and 0.1%, respectively, with higher percentages indicating superior data quality. All samples exhibited Q20 > 97.16% (threshold: >95%) and Q30 > 94.82% (threshold: >85%), with an average GC content of 43.94% (acceptable range: 40–60%). These metrics confirm the absence of sequencing bias and validate the data for downstream analysis.

### 3.2. Mapping Statistics

Unigenes were quantified using the RSEM software v1.3.3. As summarized in Table 4, unmapped reads constituted <20% of total valid reads, while uniquely mapped reads accounted for >75% (>70% threshold). Overall, >80% of reads were successfully mapped to the reference genome, demonstrating high sequencing efficiency and data utility.

### 3.3. Gene Expression Profilin

Gene expression levels, calculated as FPKM, were visualized via violin plots (Figure 1). The symmetric distribution of expression values across control and treatment groups indicated a normal distribution within the cellular population.

### 3.4. Sample Correlation Analysis

The correlation heat map (Figure 2) revealed that within the SC0 group, intra-group repeatability exceeded 0.9, while correlations with samples from the other three groups remained below 0.6. All samples across the STC1, STC4, and STC5 groups showed correlations above 0.7. Furthermore, within the STC1 and STC5 groups, intra-group correlations surpassed 0.9. Overall, this indicates a distinct difference in gene expression patterns between the control and treatment groups.

### 3.5. Inter-Group Difference Gene Statistics

Differentially expressed genes (DEGs) were identified (FDR < 0.05, |log2FC| > 1) by comparing inoculated groups (1, 4, and 5 dpi) against the 0 dpi control. Results (Figure 3) revealed 11,645 DEGs at 1 dpi (7399 upregulated,4246 downregulated), 11,285 DEGs at 4 dpi (7801 upregulated, 3484 downregulated), and 9976 DEGs at 5 dpi (7723 upregulated, 2253 downregulated). The highest number of DEGs occurred at 1 dpi, with upregulated genes consistently outnumbering downregulated genes across timepoints.

### 3.6. Venn Analysis

A total of 17,206 DEGs were identified across all timepoints (Figure 4), with 5730 shared among all groups. Pairwise comparisons revealed 1368 common DEGs between 1 dpi and 4 dpi, 547 between 1 dpi and 5 dpi, and 2225 between 4 dpi and 5 dpi.

### 3.7. Trend Analysis

Using STEM software version 1.3.13, trend analysis was performed on all differentially expressed genes from the SC0-vs.-STC1-vs.-STC4-vs.-STC5 comparisons. As shown in Figure 5, a total of 20 trend modules were generated, among which 6 were identified as statistically significant. These included Module 12 (1675 genes), exhibiting an initial stable trend followed by an increase and subsequent stabilization; Module 17 (6284 genes), showing an early increase followed by stabilization; Module 2 (1398 genes), demonstrating an initial decrease followed by stabilization; Module 19 (1069 genes), displaying a continuously increasing expression trend over time; Module 14 (968 genes), characterized by an initial increase, a mid-term decrease, and eventual stabilization; and Module 6 (608 genes), exhibiting an initial decrease followed by a subsequent increase over time. Based on the research objectives, Module 2 was selected for further emphasis.

GO enrichment analysis of the differentially expressed genes in Module 2 (top 20 enriched terms, Figure 5C) revealed significant enrichment in cellular component categories related to chloroplast and plastid structures, such as chloroplast thylakoid, plastid thylakoid, and chloroplast thylakoid membrane. Under biological processes, the genes were significantly enriched in photosynthesis. KEGG pathway enrichment analysis (top 20 pathways, Figure 5D) indicated significant enrichment in metabolic pathways including metabolic pathways, photosynthesis—antenna proteins, photosynthesis, porphyrin metabolism, starch and sucrose metabolism, and carotenoid biosynthesis.

### 3.8. GO Enrichment of Downregulated DEGs

The downregulated differentially expressed genes (DEGs) identified from comparisons between the treatment groups (inoculated with *Botrytis* strain HZ-011 for 1, 4, and 5 days) and the control group (0 days post-inoculation) were subjected to GO functional annotation. The results are as follows:

The downregulated DEGs from the comparison between the control group (inoculated for 0 days) and the treatment group inoculated with *Botrytis* strain HZ-011 for 1 day (SC0-vs.-STC1) were annotated in the GO database into three major categories: Biological Process, Molecular Function, and Cellular Component. In the Biological Process category, the largest numbers of genes were annotated to the cellular process and metabolic process, with 2096 and 1911 genes, respectively. In the Molecular Function category, the subcategories of binding and catalytic activity contained the highest numbers of annotated genes, with 1798 and 1777 genes, respectively. Within the Cellular Component category, the subcategory of cellular anatomical entity had the highest number of annotated genes, totaling 1875. These downregulated DEGs were significantly enriched in functions related to chloroplasts and plastids (Figure 6).

The downregulated DEGs from the comparison SC0-vs.-STC4 were annotated into 48 subcategories. Within the Biological Process category, the subcategories of cellular process and metabolic process contained the highest number of annotated genes, with 1499 and 1375 genes, respectively. Under the Molecular Function category, the subcategories binding and catalytic activity had the largest number of annotated genes, with 1364 and 1279, respectively. In the Cellular Component category, the subcategory cellular anatomical entity was assigned the most genes, totaling 1303. These downregulated DEGs were significantly enriched in functional terms related to chloroplast and plastid (Figure 7).

The downregulated DEGs from the comparison SC0-vs.-STC5 were annotated into 47 GO categories. Under the Biological Process category, the subcategories of cellular process and metabolic process contained the highest number of genes, with 999 and 922 annotated, respectively. Within the Molecular Function category, the subcategories binding and catalytic activity had the largest number of annotated genes, with 877 and 848, respectively. In the Cellular Component category, the subcategory of cellular anatomical entity was assigned the most genes, totaling 873. These downregulated DEGs were significantly enriched in functions associated with chloroplast and plastid (Figure 8).

### 3.9. KEGG Pathway Analysis of Downregulated DEGs

The leaves of *C. album* were inoculated with *Botrytis* strain HZ-011, and the downregulated DEGs at 1, 4, and 5 days post-inoculation were compared with those at 0 days (control) to analyze the top 30 significantly enriched KEGG pathways. The results (Figure 9) showed that metabolic pathways such as porphyrin metabolism, starch and sucrose metabolism, biotin metabolism, cyanoamino acid metabolism, thiamine metabolism, and glyoxylate and dicarboxylate metabolism; pathways related to photosynthesis including photosynthesis, photosynthesis-antenna proteins, and carbon fixation in photosynthetic organisms; pathways of secondary metabolite biosynthesis such as carotenoid biosynthesis, anthocyanin biosynthesis, ubiquinone and other terpenoid–quinone biosynthesis, sesquiterpenoid and triterpenoid biosynthesis, and indole alkaloid biosynthesis, as well as plant circadian rhythm, pentose and glucuronate interconversions, and glycosaminoglycan degradation were consistently enriched across all three comparison groups. This indicates that these pathways play important roles in the response of *C. album* to infection by *Botrytis* strain HZ-011. In addition to these common pathways, unique pathways were identified in each treatment group compared with the control. Specifically, in the SC0-vs.-STC1 group, base excision repair, DNA replication, and biosynthesis pathways of stilbenoids, diarylheptanoids and gingerols, isoquinoline alkaloids, flavonoids, and phenylpropanoids were exclusively enriched. The unique pathways in SC0-vs.-STC4 included glycine, serine and threonine metabolism, pyrimidine metabolism, protein export, and ABC transporters. In SC0-vs.-STC5, unique pathways consisted of terpenoid backbone biosynthesis and phagosomes. Notably, the photosynthesis pathway was significantly enriched among the downregulated DEGs in all three comparison groups. In summary, the KEGG enrichment analysis of the downregulated DEGs was closely associated with plant photosynthesis and essential metabolic processes, indicating that infection by *Botrytis* suppressed pathways was related to the synthesis of defense compounds and the regulation of photosynthesis in *C. album*.

### 3.10. Photosynthesis-Related Metabolic Pathway Analysis

Based on the KEGG enrichment analysis results, the photosynthetic metabolic pathways were further analyzed. A total of 32, 21, and 7 unigenes were annotated to the significantly enriched pathways of photosynthesis, photosynthesis—antenna proteins (light reaction), and carotenoid biosynthesis, respectively (Figure 10).

The significantly downregulated genes annotated to the photosynthesis pathway included *PSAE-1* (Unigene0000564), *PSAF* (Unigene0016767), *PSAL* (Unigene0090037), *PSAK* (Unigene0002170), *PSBQ* (Unigene0029403), *PSAN* (Unigene0002771), *PSBP* (Unigene0090986), *PSBO* (Unigene0012369), *PSBY* (Unigene0013148), *PSAO* (Unigene0062481), *PSAG* (Unigene0003221), *PPL1* (Unigene0005962), *SB27-1* (Unigene0016630), *PSB28* (Unigene0020151), *PNSL2* (Unigene0024271), *ATPC* (Unigene0089356), *PSBS* (Unigene0093649), *PSAN* (Unigene0002770), *PSAH* (Unigene0000860), *FDC2* (Unigene0000883), *PSBW* (Unigene0032465), *PETH* (Unigene0093448), *PSBS* (Unigene0093650), *PETE* (Unigene0073990), *ATPD* (Unigene0034986), and *ATPF2* (Unigene0077804). These genes encode the following key photosynthetic components: F-type H^+^-transporting ATPase subunit b, ATPF1D-atpH, F-type H^+^/Na^+^-transporting ATPase subunit gamma, Plastocyanin, Ferredoxin, Ferredoxin—NADP^+^ reductase, Photosystem I subunit II, Photosystem I subunit IV, Photosystem I subunit III, Photosystem I subunit VI, Photosystem I subunit X, Photosystem I subunit XI, Photosystem I subunit PsaN, Photosystem II CP43 chlorophyll apoprotein, Photosystem II oxygen-evolving enhancer protein 1, Photosystem II oxygen-evolving enhancer protein 2, Photosystem II PsbW protein, Photosystem II PsbY protein, Photosystem II 22 kDa protein, Photosystem II oxygen-evolving enhancer protein 3, Photosystem II Psb27 protein, Photosystem II 13 kDa protein, Photosystem I subunit V, and Photosystem I subunit PsaO.

The significantly downregulated genes annotated to the photosynthesis—antenna proteins (light reaction) pathway included *LHCA6* (Unigene0006338), *LHCB4.1* (Unigene0027841), *LHCA3* (Unigene0065201), *CAB7* (Unigene0020504), *CAP10A* (Unigene0031622), *1hcA-P4* (Unigene0024358), *LHCA5* (Unigene0034988), *CAB7* (Unigene0020303), *CAB36* (Unigene0058827), *LHCB5* (Unigene0014874), *LHCA1* (Unigene0091487), *LHCA4* (Unigene0024360), *CAB91R* (Unigene0053709), *CAP10A* (Unigene0031621), *1hcA-P4* (Unigene0024359), *LHCB4.2* (Unigene0092477), *CAB13* (Unigene0027733), *CAB40* (Unigene0020302), *CAB91R* (Unigene0054725), *LHCB4.2* (Unigene0064245), and *CAB91R* (Unigene0027734). These genes encode the following light-harvesting chlorophyll a/b-binding proteins (LHC): Light-harvesting complex I chlorophyll a/b-binding protein 1, Light-harvesting complex I chlorophyll a/b-binding protein 2, Light-harvesting complex I chlorophyll a/b-binding protein 3, Light-harvesting complex I chlorophyll a/b-binding protein 4, Light-harvesting complex I chlorophyll a/b-binding protein 5, Light-harvesting complex II chlorophyll a/b-binding protein 1, Light-harvesting complex II chlorophyll a/b-binding protein 2, Light-harvesting complex II chlorophyll a/b-binding protein 3, Light-harvesting complex II chlorophyll a/b-binding protein 4, Light-harvesting complex II chlorophyll a/b-binding protein 5, and Light-harvesting complex II chlorophyll a/b-binding protein 6 (Figure 11).

The significantly downregulated genes annotated to the carotenoid biosynthesis pathway included *ZDS* (Unigene0018609), *CRTL-E-1* (Unigene0024093), *CYP97A3* (Unigene0025522), *SOVF_077800* (Unigene0094460), *CYP97C1* (Unigene0090532), *CRTISO* (Unigene0063335), and *CCD4* (Unigene0091209). These genes encode the following key enzymes: zeta-carotene desaturase, lycopene epsilon-cyclase, prolycopene isomerase, carotenoid epsilon hydroxylase, zeaxanthin epoxidase, 9-cis-epoxycarotenoid dioxygenase, and beta-ring hydroxylase, respectively. Furthermore, four potential target genes—*PSB28*, *PSBP*, *CAP10A*, and *CRTL-E-1*—were identified as being significantly affected by the *Botrytis* strain HZ-011 infection. These genes are hypothesized to play important roles in the weed-inhibiting process facilitated by the pathogen (Figure 12).

### 3.11. Validation of Differentially Expressed Genes in the Transcriptome by qRT-PCR

Based on the transcriptomic data, 20 differentially expressed genes were selected for qRT-PCR experiments and analysis. The relative expression levels of the genes, calculated using FPKM and qRT-PCR results, were plotted on the vertical axis, while the treatment timepoints were plotted on the horizontal axis to create a dual-*y*-axis bar-line chart. As shown in Figure 13, the trends in FPKM values over time for the selected 20 differentially expressed genes were generally consistent with the trends in relative expression levels calculated from the qRT-PCR results.

## 4. Discussion

Transcriptome sequencing technology has become a crucial tool in studying plant–microbe interaction mechanisms. Its high-throughput and high-precision advantages enable a better revelation of molecular biological changes in the microscopic world, providing reliable support for the improvement in biological control technologies [23,24]. Currently, RNA-Seq technology has been widely applied in studies of plant–microbe interactions, paving the way for identifying key metabolic pathways and genes involved in plant–pathogen interactions [25]. For instance, Zhao Q [26], aiming to clarify the pathogenic mechanism of gummy stem blight (*Mycosphaerella melonis*) in seed-used pumpkin, conducted a comparative transcriptome analysis between pumpkin leaves inoculated with *Stagonosporopsis cucurbitacearum* and a PDA-inoculated control group. A total of 319 DEGs were identified, including 129 upregulated and 190 downregulated genes. KEGG annotation results indicated seven pathways related to pathogenicity, with 86 DEGs involved in metabolite synthesis, suggesting that metabolic or biosynthetic pathways were closely associated with the strain’s pathogenicity. In another study, Zhang Y et al. [27] used transcriptome sequencing to investigate the interaction mechanism between *Verticillium dahliae* and cotton (*Gossypium hirsutum*). The sequencing yielded 46,192 high-quality expressed sequences, and 3027 differentially expressed genes were annotated in the KEGG database. These DEGs included a large number of transcripts related to cotton defense, implying that the accumulation of these defense transcripts may be associated with cotton resistance to Verticillium wilt. These findings provide a theoretical basis for resistance breeding in cotton. In this study, transcriptome sequencing was employed to analyze gene expression changes in *C. album* during infection by *Botrytis* strain HZ-011. The treatment groups consisted of samples inoculated for 1, 4, and 5 days, compared with a control group (0 days), to screen for significantly differentially expressed genes in response to the infection. The results showed that at 1 day post-inoculation, there were 11,645 DEGs, with 7399 upregulated and 4246 downregulated genes; at 4 days, 11,285 DEGs were identified, including 7801 upregulated and 3484 downregulated genes; and at 5 days, 9976 DEGs were detected, with 7723 upregulated and 2253 downregulated genes. Over time, the upregulation of genes first increased and then decreased and downregulated genes decreased, indicating that the infectious activity of *Botrytis* strain HZ-011 continued to intensify, while the defense capacity of *C. album* gradually weakened under biotic stress.

Photosynthesis serves as the primary source of nutrients during plant growth. Through their leaves and chlorophyll, plants absorb sunlight to perform photosynthesis, and the resulting organic compounds are transported to various parts of the plant via the vascular system, ensuring nutritional supply and normal growth. GO functional analysis of the downregulated differentially expressed genes (DEGs) in this study revealed significant enrichment in functions related to chloroplasts and plastids. KEGG enrichment results further demonstrated that, following infection by *Botrytis* strain HZ-011, the downregulated genes in *C. album* were markedly enriched in pathways associated with photosynthesis, including the photosynthesis pathway, photosynthesis—antenna proteins pathway, and carotenoid biosynthesis.

Inhibition of photosynthesis directly leads to energy depletion. Without sufficient ATP and sugars, plants struggle to sustain basic metabolism, let alone support energy-intensive defense responses; growth halts, leaves yellow, and ultimately, the plant dies due to ineffective defense mechanisms [28,29]. Dysfunction in the antenna protein pathway reduces energy capture efficiency and increases photooxidative damage [30,31]. Plants not only fail to acquire energy but expend additional energy repairing chloroplast proteins and membrane systems damaged by ROS, further exacerbating the energy crisis. Disruption of the carotenoid pathway disables the plant’s “sun protection system,” leaving photosynthetic organs unprotected under stress and exposed to light [32,33]. This accelerates energy collapse and cell death. In the interaction between the *Botrytis* strain HZ-011 and *C. album*, these three pathways are affected synergistically, trapping plants in a vicious cycle of photosynthetic organ damage–energy depletion death cycle. By directly targeting or indirectly inducing stress at various stages of energy production, the *Botrytis* strain HZ-011 ultimately depletes the energy reserves of *C. album*, weakening its defense capabilities and thereby successfully infecting and causing its death. This aligns with findings by Cheng et al. [34], who observed significant suppression of photosynthesis-related gene expression in *C. album* leaves inoculated with *Aureobasidium pullulans* PA-2. Thus, *Botrytis* strain HZ-011 achieves weed suppression by disrupting photosynthetic processes, leading to the sustained downregulation of key genes and metabolites along these metabolic pathways. Understanding this mechanism provides important guidance for future efforts to achieve fungal weed control through genetic engineering and provides direct molecular tools and insights for the development and optimization of *Botrytis* strain HZ-011 in practical agricultural applications. This also confirms that compared with chemical herbicides, microbial herbicides possess multi-target characteristics that disrupt weed physiology by producing specific toxins or enzymes. Their complex mode of action makes them less prone to resistance development [6,35]. The ideal weed management strategy does not simply involve replacing chemical herbicides with biological alternatives. Instead, it requires the integrated use of microbial herbicides, low-risk chemical herbicides, and agroecological practices (such as crop rotation and mulching) [36,37]. This approach aims to effectively control weeds while minimizing negative impacts on the environment and human health, ultimately achieving sustainable agricultural development.

## 5. Conclusions

This study represented the first application of high-throughput RNA-seq technology to investigate the mechanism of *Botrytis* infection in *C. album*. The gene expression profiles of *C. album* at three different timepoints after inoculation with *Botrytis* strain HZ-011 were systematically analyzed. One day after inoculation with the *Botrytis* strain HZ-011, a total of 11,645 differentially expressed genes were identified, including 7399 upregulated genes and 4246 downregulated genes. Four days post-inoculation, 11,285 differentially expressed genes were detected, comprising 7801 upregulated genes and 3484 downregulated genes. After 5 days of inoculation, 9976 differentially expressed genes were identified, comprising 7723 upregulated genes and 2253 downregulated genes. Following infection by *Botrytis* strain HZ-011, downregulated genes in *C. album* were significantly enriched in pathways associated with photosynthesis, including the photosynthesis pathway, the photosynthesis—antenna protein pathway, and the carotenoid biosynthesis pathway. Based on database annotations and the expression trends of differentially expressed genes, four potential target genes—*PSB28*, *PSBP*, *CAP10A*, and *CRTL-E-1*—affected by *Botrytis* strain HZ-011 were identified. Further functional analysis and validation of these candidate genes will be conducted to elucidate the molecular basis of the herbicidal activity of strain HZ-011 and provide potential targets for the development of microbial herbicides. This study presented the transcriptomic research on the response of *C. album* to infection by *Botrytis* strain HZ-011 but did not address potential correlations with protein or metabolite levels. Further research will be conducted to validate the herbicidal mechanism through proteomics or metabolomics analysis. This will lay the foundation for elucidating the herbicidal activity of the HZ-011 strain and provide targets for the development of microbial herbicides.

## Figures and Tables

**Figure 1 microorganisms-13-02177-f001:**
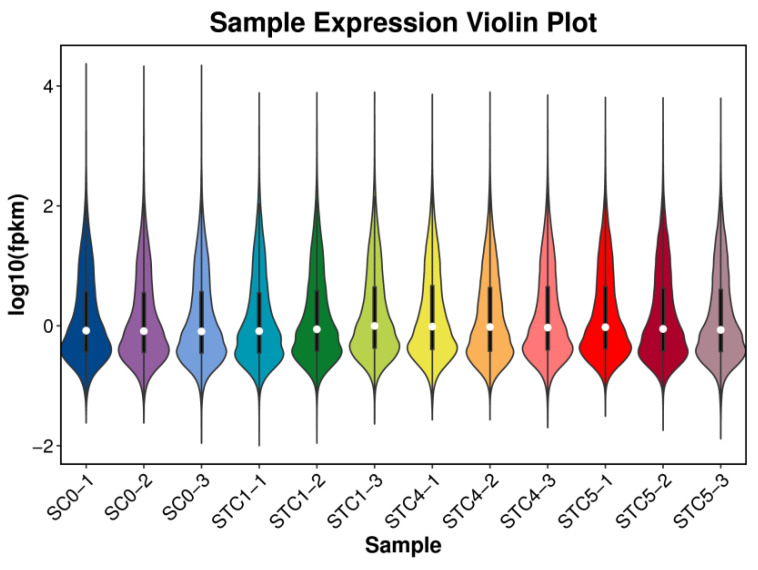
Gene expression violin plot.

**Figure 2 microorganisms-13-02177-f002:**
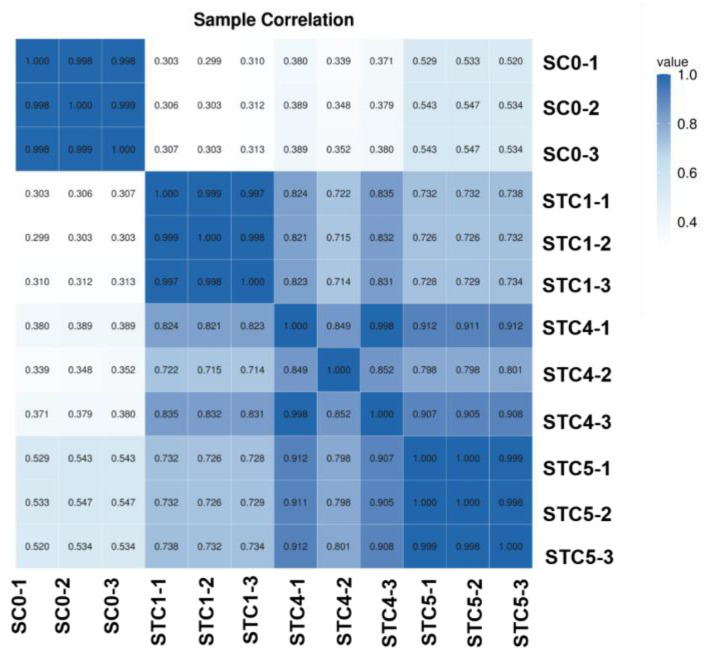
Sample and correlation heat map.

**Figure 3 microorganisms-13-02177-f003:**
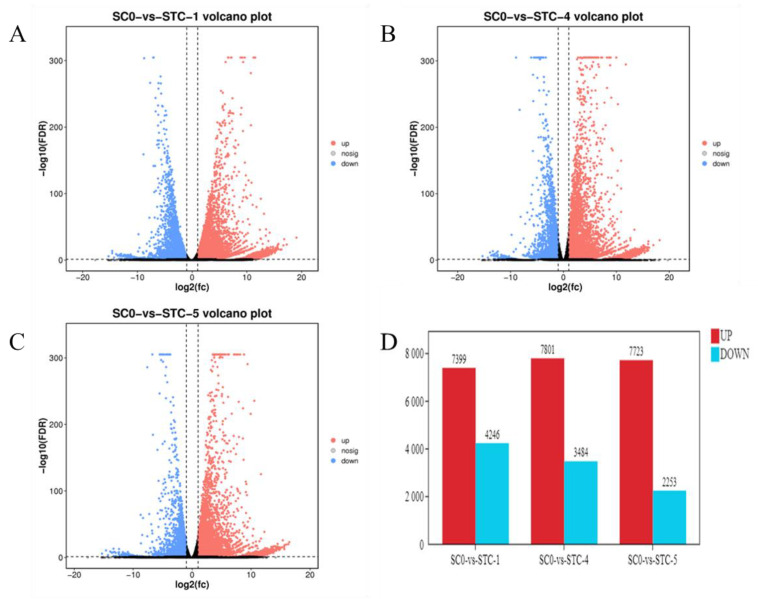
Differentially expressed gene statistics. (**A**): SCO-vs.-STC1; (**B**): SC0-vs.-STC4; (**C**): SC0-vs.-STC5; (**D**): statistical graph of the number of differentially expressed genes.

**Figure 4 microorganisms-13-02177-f004:**
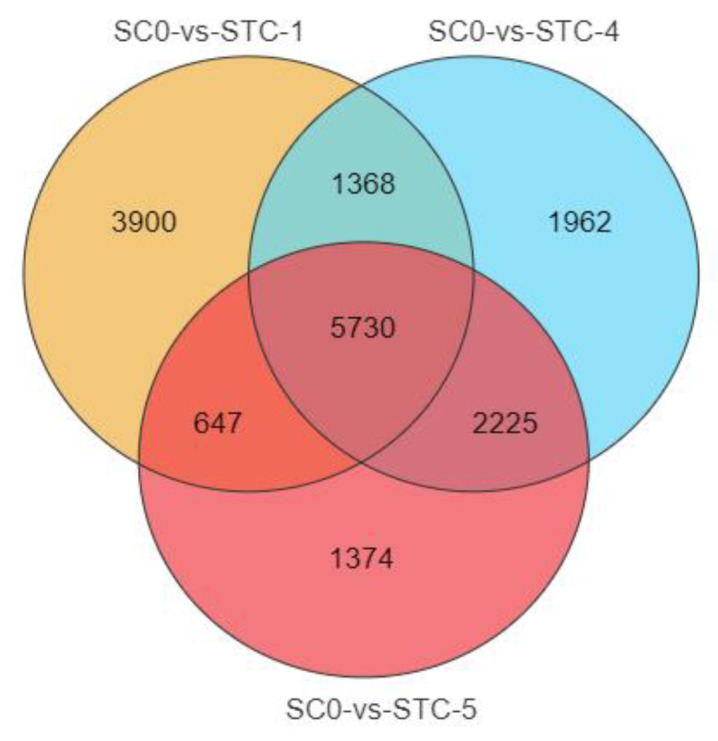
Venn diagram of differences between treatments and control. orange: SCO-vs-STC-1; blue: SCO-vs-STC-4; red: SCO-vs-STC-5.

**Figure 5 microorganisms-13-02177-f005:**
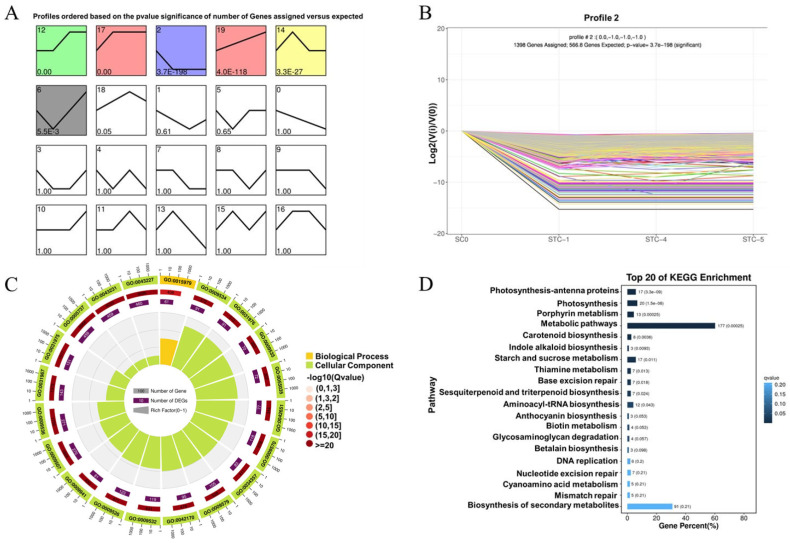
Differential gene trend analysis. (**A**): Total graph of all trends (*p*-value sorted); (**B**): trend graph of Module 2 differential genes; (**C**): circle graph of Module 2 differential gene GO enrichment analysis; (**D**): bar graph of module 2 differential gene KEGG enrichment analysis.

**Figure 6 microorganisms-13-02177-f006:**
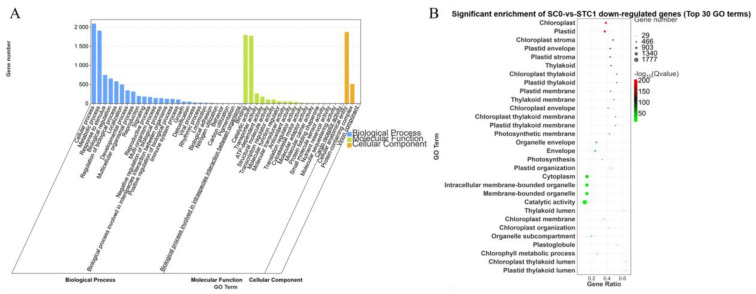
GO enrichment analysis of SC0-vs.-STC1 downregulated differential genes. (**A**): SC0-vs.-STC1 downregulation of DEGs GO-enriched secondary classification; histogram (**B**): SC0-vs.-STC1 downregulation of DEGs GO significantly enriched bubble plot.

**Figure 7 microorganisms-13-02177-f007:**
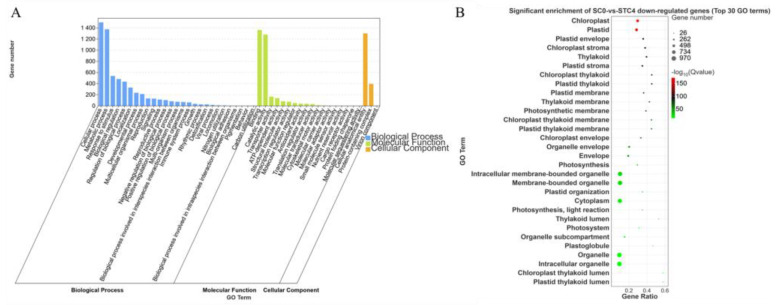
GO enrichment analysis of SC0-vs.-STC4 downregulated differential genes. (**A**): SC0-vs.-STC4 downregulation of DEGs GO-enriched secondary classification; histogram (**B**): SC0-vs.-STC4 downregulation of DEGs GO significantly enriched bubble plot.

**Figure 8 microorganisms-13-02177-f008:**
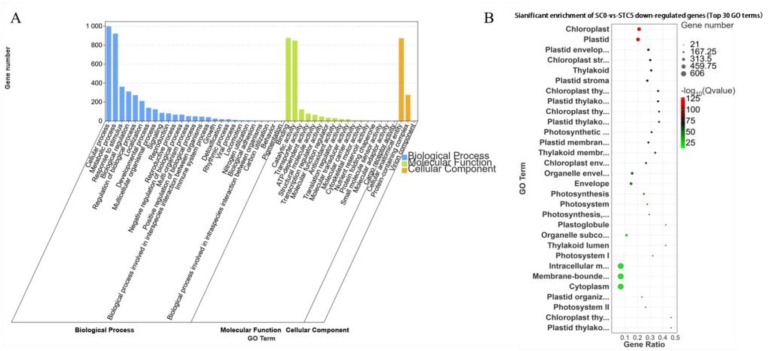
GO enrichment analysis of SC0-vs.-STC5 downregulated differential genes. (**A**): SC0-vs.-STC5 downregulation of DEGs GO-enriched secondary classification histogram (**B**): SC0-vs.-STC5 downregulation of DEGs GO significantly enriched bubble plot.

**Figure 9 microorganisms-13-02177-f009:**

Enrichment analysis of KEGG pathway for downregulated differentially expressed genes (top 30 pathways). (**A**) SC0-vs.-STC1 downregulation of DEGs KEGG pathway significantly enriched bubble plot; (**B**) SC0-vs.-STC4 downregulation of DEGs KEGG pathway significantly enriched bubble plot; (**C**) SC0-vs.-STC5 downregulation of DEGs KEGG pathway significantly enriched bubble p1lot.

**Figure 10 microorganisms-13-02177-f010:**
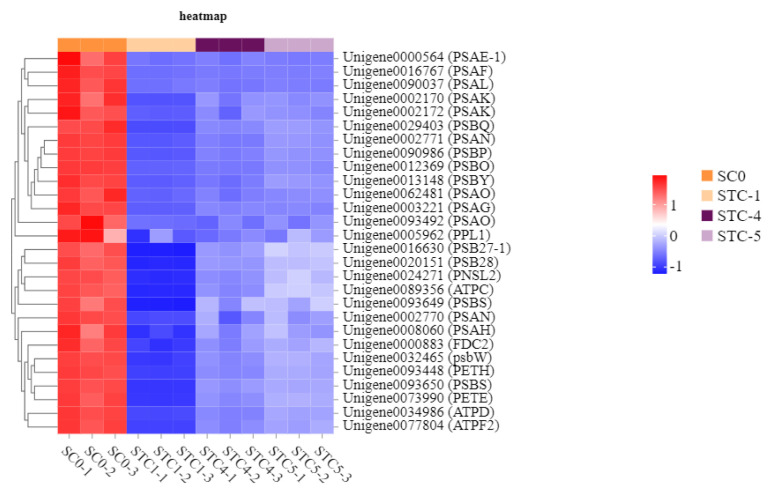
Heat map of the expression of genes regulated by the photosynthesis pathway.

**Figure 11 microorganisms-13-02177-f011:**
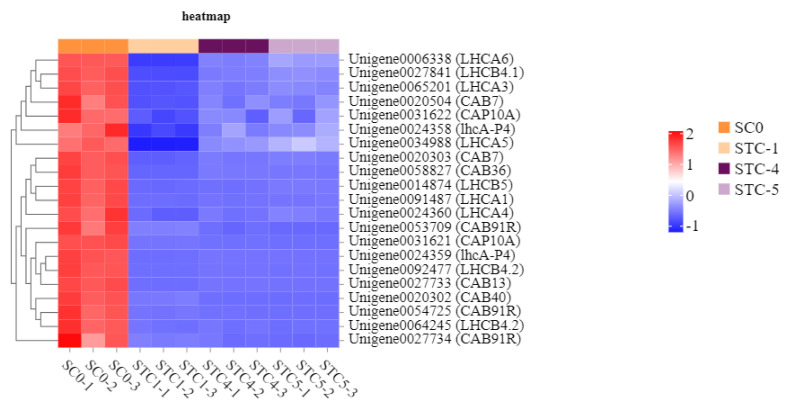
Heat map of the amount of gene expression regulated by the photosynthesis—antenna protein pathway.

**Figure 12 microorganisms-13-02177-f012:**
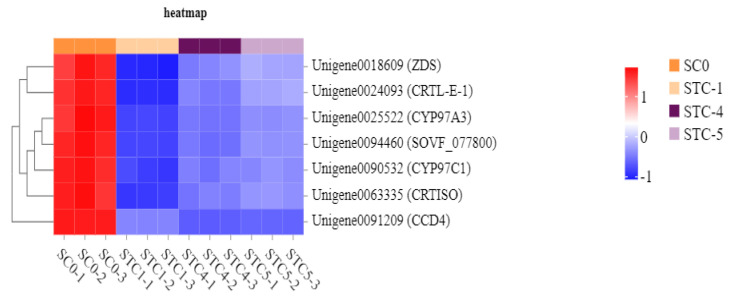
Heat map of carotenoid pathway-regulated gene expression.

**Figure 13 microorganisms-13-02177-f013:**
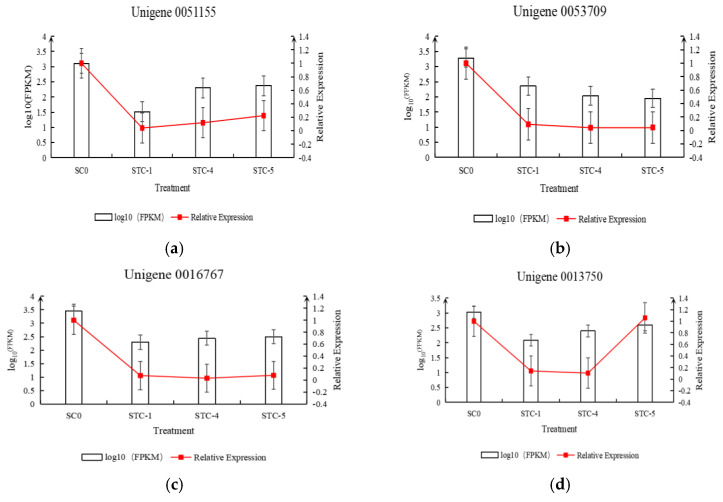
Trend of differential gene qRT-PCR and FPKM changes.

**Table 1 microorganisms-13-02177-t001:** The qRT-PCR reaction system.

Reagents	Dosage (μL)
2× FastReal qPCR PreMix	12.5 μL
PCR Forward Primer (10 μM)	1.0 μL
PCR Reverse Primer (10 μM)	1.0 μL
cDNA	1 μL
50× ROX Reference Dye	1 μL
ddH_2_O up to	25 μL

**Table 2 microorganisms-13-02177-t002:** Sequence of primers for qRT-PCR of the transcriptome.

Unigene ID	Forward Primer (5′→3′)	Reverse Primer (5′→3′)	Size
Unigene 0058827	GATGTGTATTCCCTGAAATCCTGTCC	TTCCTAAGTAGTCGAGTCCACCTTC	111
Unigene 0031621	TTACCTCGCCCTTCGTGTCG	CAGCAGCCACAACCAACTTCC	117
Unigene 0024359	ACAGAAGCACATAAGCGATTGAGC	GTGTAAGCGGATTTGGTTAAGTGATTC	117
Unigene 0093448	GCTTGGCTTGGCTCTTCTTGG	AGCGAAATCTAGTCTGAAGTTGTCTG	110
Unigene 0073990	GCCGTCGCCGTCCCATC	GGATGCCTTGACTGACATTCTTGG	114
Unigene 0069442	CGAATACGATGAGCAGGACAACTAC	GCAGCCACGGCATTGAAGAG	117
Unigene 0051155	CGCTGCCGCTATTTCTCCTG	TTCCATTGTTGCTGCCCTCTTG	119
Unigene 0092477	GATGATGCTCCTGTGGTCTTATGTG	CTGAAACCTTGTCTCTGTCTACTTCTG	118
Unigene 0008059	AGTGGTTGCTAAGTATGGTGACAAG	CTGAAGAGAGTTGTAAGGTGAAGGAG	118
Unigene 0014874	CTCCTTGGTGGTGCTGAATACTAC	GCTGCTTGGTCTGGGTCATTG	116
Unigene 0020154	CGCCACTGCTGCTGCTATTG	CTGCCTCGTCCAATCTCAAGTG	120
Unigene 0057427	GGTAACGAGACTGCTGGTTCATAC	TCAATGGATGTTCCTGGCAAGTTAG	110
Unigene 0052136	CGGTCCATCAATAAGGCGTGTAG	ATCCTCCGCTAGTAATTTCACCAATG	113
Unigene 0012369	GCATCAACAACCTTCTTACAACAACC	ACTCTACCAGCACCAGAATCTACAC	119
Unigene 0062583	CCTCATCTGGGCTGGGCTTAG	CCTACTCTAATGGCTGCTTTCTCTG	115
Unigene 0089933	TGCGGACATCATTCAGGTTTCAG	GGGTCTGGTTGGTTTCTGTAAGTC	116
Unigene 0067080	GCAGCCGAGACTACCGAGAC	TCCTCAATAACCCACCTGTGCTAC	115
Unigene 0013750	GCAAGCCTATCAAGGTTGTCTCTG	TGCCAGCACCAGGACCATC	117
Unigene 0016767	CATCAAAGATAGTATGCCAGCAACAAG	GGACAGTAGCACCGAGGATAGAG	113
Unigene 0053709	GAGATTAAGAACGGTAGGTTGGCTATG	GGGTCGGCAAGGTGGTCAG	110
Actin	CACGGCTTACTGGAGGAATGAAGTC	CTCAGGAGTTGAAGCAAGTACGGATC	113

**Table 3 microorganisms-13-02177-t003:** Sequencing data quality statistics.

Sample	Raw Data	Clean Reads	Clean Bases (bp)	Q20 (%)	Q30 (%)	GC (%)
SC0-1	39,919,152	39,711,448	5,899,769,754	97.16%	94.84%	44.66%
SC0-2	41,376,190	41,174,526	6,122,274,191	97.17%	94.83%	44.60%
SC0-3	47,152,598	46,930,324	6,968,056,372	97.17%	94.82%	44.68%
STC1-1	49,926,622	49,769,036	7,370,058,700	98.18%	96.65%	43.41%
STC1-2	44,475,938	44,273,224	6,501,022,117	98.03%	96.31%	43.36%
STC1-3	36,015,926	35,866,336	5,303,369,524	98.10%	96.43%	43.37%
STC4-1	40,086,282	39,923,870	5,865,940,888	98.14%	96.53%	43.78%
STC4-2	48,348,462	48,172,514	7,162,947,827	98.29%	96.75%	44.31%
STC4-3	41,019,236	40,876,836	6,007,706,583	98.19%	96.62%	43.77%
STC5-1	40,641,004	40,469,252	5,985,931,564	98.20%	96.57%	43.84%
STC5-2	47,536,780	47,355,332	7,018,085,006	98.21%	96.63%	43.75%
STC5-3	46,767,836	46,594,110	6,902,075,112	98.23%	96.64%	43.74%
Average	43,605,502	43,426,401	6,425,603,137	97.92%	96.14%	43.94%
Total	523,266,026	521,116,808	77,107,237,638			

Note: SC0 is the control group; STC1, STC4, and STC5 are the treatment groups inoculated for 1 d, 4 d, and 5 d, respectively.

**Table 4 microorganisms-13-02177-t004:** Unigene comparison rate statistics.

Sample	Total	Unmapped (%)	Unique—Mapped (%)	Total—Mapped (%)
STC0-1	39,711,448	7,557,650 (19.03%)	30,892,390 (77.79%)	32,153,798 (80.97%)
STC0-2	41,174,526	7,783,910 (18.90%)	32,100,278 (77.96%)	33,390,616 (81.10%)
STC0-3	46,930,324	8,766,495 (18.68%)	36,637,247 (78.07%)	38,163,829 (81.32%)
STC1-1	49,769,036	8,095,692 (16.27%)	39,509,100 (79.38%)	41,673,344 (83.73%)
STC1-2	44,273,224	7,311,350 (16.51%)	35,103,000 (79.29%)	36,961,874 (83.49%)
STC1-3	35,866,336	6,028,367 (16.81%)	28,368,882 (79.10%)	29,837,969 (83.19%)
STC4-1	39,923,870	6,866,114 (17.20%)	31,175,109 (78.09%)	33,057,756 (82.80%)
STC4-2	48,172,514	8,280,096 (17.19%)	37,938,523 (78.76%)	39,892,418 (82.81%)
STC4-3	40,876,836	6,974,451 (17.06%)	31,953,261 (78.17%)	33,902,385 (82.94%)
STC5-1	40,469,252	6,651,150 (16.44%)	32,073,452 (79.25%)	33,818,102 (83.56%)
STC5-2	47,355,332	7,825,710 (16.53%)	37,526,212 (79.24%)	39,529,622 (83.47%)
STC5-3	46,594,110	7,594,067 (16.30%)	36,997,679 (79.40%)	39,000,043 (83.70%)

## Data Availability

The original contributions presented in this study are included in the article. Further inquiries can be directed at the corresponding author.

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
