# Peer review of "Transcriptome Analysis of Chenopodium album in Response to Infection by Botrytis Strain HZ-011"

_microorganisms, 2025, doi:10.3390/microorganisms13092177_

Round 1
Reviewer 1 Report
Comments and Suggestions for Authors
Dear Authors,
After a careful review of your manuscript, I would like to share some constructive comments and suggestions that may help strengthen your work:
Abstract:
- The abstract doesn’t clearly state what is novel compared to previous studies.
- Why Chenopodium album was chosen (weed management relevance?).
- Why strain HZ-011 is important (unique herbicidal potential?).
- Were early responses stronger than later ones?
- Which biological processes dominated across timepoints?
Introduction:
- The mention of Qinghai Province makes the problem seem very location-specific, while C. album is a global weed problem.
- Please provide previous transcriptomic or molecular studies in weed–pathogen interactions (if any exist).
- It would be stronger if the authors link Botrytis pathogenicity traits directly to potential herbicidal effects on weeds.
- The introduction is lengthy, please be concise to the topic.
Materials and Methods:
1. Spore Suspension Preparation
• The description of scraping, filtering, centrifuging, and diluting is overly narrative and could be simplified into a stepwise protocol.
• It is unclear at what centrifugation speed and time the suspension was processed. Without this, reproducibility suffers.
2. Inoculation Procedure
• Inoculation is described as both spray and injection — but the ratio or method is not specified clearly. Were both methods applied to the same plants or to different groups?
• 25 mL per pot seems like a large volume. The rationale should be explained.
3. Biological replicates: Were these independent plants or pooled samples? And were technical replicates also performed?
4. Library prep: Which kit was used for library construction? Insert size? Paired-end or single-end sequencing? Read length?
5. Were genes selected randomly, or based on specific pathways?
6. Why was Actin chosen as the reference gene? Its stability under pathogen stress is questionable, validation of housekeeping genes is expected.
7. qPCR was performed in a 50 μL reaction, this is unusually large (standard is 10–20 μL). Why was such a large reaction volume used?
Results:
- Sample correlation analysis: Reporting r = 1 for all triplicates is highly suspicious, biological replicates are never perfectly correlated. This suggests either a plotting error or insufficient biological variability. Needs re-checking.
- Instead of only showing Pearson correlation, PCA plots or hierarchical clustering would better demonstrate reproducibility and treatment separation.
- Figure 4: check the spelling for Venn diagram.
- STEM trend analysis is reported, but only Module 2 is discussed. Why not describe other significant modules? Focusing on one looks selective unless justified.
Discussion:
- Provide a more detailed comparison of your results with those studies.
- How does Botrytis HZ-011’s suppression of photosynthesis compare with the strategies of other biocontrol fungi?
- Discuss specific functional categories of upregulated DEGs (e.g., stress response, secondary metabolism, hormone signaling).
- Interpret whether these upregulations represent fungal manipulation of host processes or residual host defense attempts.
- Expand on how suppression of antenna proteins, carotenoid biosynthesis, and PSI/PSII components directly impacts energy metabolism and plant survival.
- Discuss how using Botrytis HZ-011 as a microbial herbicide compares with chemical herbicides in terms of sustainability, resistance risks, and safety.
- Acknowledge potential limitations.
Conclusion:
Please add a conclusion section.
Author Response
Dear revier 1,
Thank you very much for your comments and valuable suggestions on our manuscript, we have carefully made major revisions according to your comments. All changes have been clearly highlighted in the submited manuscript.
Below is our point by point response to these comments:
After a careful review of your manuscript, I would like to share some constructive comments and suggestions that may help strengthen your work:
Response: The reviewer clearly possessed profound knowledge and a broad perspective in this field. The questions you raised were both comprehensive and incisive, helped us refine the manuscript from multiple angles. We have addressed each point, and we believe the revised manuscript has been significantly enhanced. Thank you for dedicating your time and effort to help improve our work.
Abstract:
- The abstract doesn’t clearly state what is novel compared to previous studies.
Response: Thank you for pointing this out. The novelty of this study lies in the High-throughput RNA-seq technology to analyze the transcriptomes of C. album leaves post-inoculation with Botrytis strain HZ-011, which has mentioned in the introduction.
- Why Chenopodium album was chosen (weed management relevance?).
Response: Strain HZ-011 exhibited highly effective herbicidal activity against various broadleaf weeds, with the highest control efficacy observed against Chenopodium album ( Reference 21).
- Why strain HZ-011 is important (unique herbicidal potential?).
Response: Botrytis strain HZ-011, isolated from the Qinghai-Tibet Plateau, exhibited potent herbicidal activity against C. album and other weeds. It is an important microbial resource.
- Were early responses stronger than later ones?
Response: Yes, the reaction is from strong to weak.
- Which biological processes dominated across time points?
Response: The dominant biological processes across time points have been supplemented.
Introduction:
- The mention of Qinghai Province makes the problem seem very location-specific, while C. album is a global weed problem.
Response: Agree. C. album is a pervasive and highly detrimental weed worldwide. This study used C. album from Qinghai Province as experimental material, hence the introduction section briefly mentioned the harm caused by C. album in this region.
- Please provide previous transcriptomic or molecular studies in weed–pathogen interactions (if any exist).
Response: This section has been supplemented.
- It would be stronger if the authors link Botrytis pathogenicity traits directly to potential herbicidal effects on weeds.
Response: This study employed high-throughput RNA-seq technology to perform transcriptome sequencing on leaves of C. album inoculated with Botrytis strain HZ-011 at 1d, 4d, and 5d post-inoculation. The primary focus is on C. album 's transcriptome. Regarding the pathogenic mechanism of Botrytis pathogenicity you proposed, we have completed the transcriptome sequencing of Botrytis and identified regulatory genes associated with the pathogenicity of this strain. This part will be presented in a separate manuscript.
- The introduction is lengthy, please be concise to the topic.
Response:The introduction section has been streamlined.
Materials and Methods:
- Spore Suspension Preparation
- The description of scraping, filtering, centrifuging, and diluting is overly narrative and could be simplified into a stepwise protocol.
- It is unclear at what centrifugation speed and time the suspension was processed. Without this, reproducibility suffers.
Response: Agree.This section has been simplified and supplemented as your request.
- Inoculation Procedure
- Inoculation is described as both spray and injection — but the ratio or method is not specified clearly. Were both methods applied to the same plants or to different groups?
- 25 mL per pot seems like a large volume. The rationale should be explained.
Response: These two methods are standard approaches for inoculating plants with pathogens. 25 mL per pot is sufficient to thoroughly moisten the entire leaf surface.
- Biological replicates: Were these independent plants or pooled samples? And were technical replicates also performed?
Response: These were pooled samples, technical replicates were also performed.
- Library prep: Which kit was used for library construction? Insert size? Paired-end or single-end sequencing? Read length?
Response: This section has been supplemented.
- Were genes selected randomly, or based on specific pathways?
Response: Differentially expressed genes with significant changes were selected.
- Why was Actin chosen as the reference gene? Its stability under pathogen stress is questionable, validation of housekeeping genes is expected.
Response: Your suspicion is entirely correct. Based on our transcriptome sequencing data, we should first screen multiple candidate internal reference genes. After reviewing the literature, we selected the commonly used internal reference gene Actin for qRT-PCR. However, the stability of Actin in plant-pathogen interaction studies requires further validation. We will certainly pay attention to this in the future.
- qPCR was performed in a 50 μL reaction, this is unusually large (standard is 10–20 μL). Why was such a large reaction volume used?
Response: This section has been modified.. In our experiment, both 25 μL and 50 μL reaction have been performed. The reason for the large reaction volume is that the purpose of qPCR is not only quantification, but also the plan to recover PCR products for subsequent cloning and sequencing.
Results:
- Sample correlation analysis: Reporting r = 1 for all triplicates is highly suspicious, biological replicates are never perfectly correlated. This suggests either a plotting error or insufficient biological variability. Needs re-checking.
Response:Thank you very much for your valuable feedback. We take your concern regarding the sample correlation r values very seriously and immediately re-examined the raw data. Upon careful review, we found that the textual description in the results section was not sufficiently precise. The exact values are 1. We apologize for any confusion this may have caused. However, the data itself is entirely accurate, as demonstrated in the figure: the biological replicates within the SC0 group exhibit extremely high correlation, with values of 0.998, 0.999, and 1.000 (e.g., r = 0.998 between SCO-1 and SCO-2, r = 0.999 between SCO-2 and SCO-3). While exceptionally high, this does not reach perfect 1, indicating our experiments exhibit extremely high technical reproducibility and minimal biological variability—a phenomenon possible in well-controlled experiments. Intra-group repeatability within the STC5 group was similarly exceptionally high, reaching 0.998, 0.999, and 1.000 (e.g., r = 1.000 between STC5-1 and STC5-2, r = 0.998 between STC5-2 and STC5-3). More importantly, as previously noted, correlations between the SC0 group and other treatment groups (STC1, STC4, STC5) were all below 0.6, while inter-sample correlations within treatment groups (STC1, STC4, STC5) exceeded 0.7. The descriptions in the manuscript have been revised.
- Instead of only showing Pearson correlation, PCA plots or hierarchical clustering would better demonstrate reproducibility and treatment separation.
Response:This section has been supplemented.
- Figure 4: check the spelling for Venn diagram.
Response: The spelling has been corrected.
- STEM trend analysis is reported, but only Module 2 is discussed. Why not describe other significant modules? Focusing on one looks selective unless justified.
Response: Descriptions for other significant modules have been supplemented.
Discussion:
- Provide a more detailed comparison of your results with those studies.
Response: This section has been supplemented.
- How does Botrytis HZ-011’s suppression of photosynthesis compare with the strategies of other biocontrol fungi?
Response: There is very little literature on Botrytis HZ-011’s suppression of photosynthesis compare with the strategies of other biocontrol fungi, and we have analyzed it in our discussion.
- Discuss specific functional categories of upregulated DEGs (e.g., stress response, secondary metabolism, hormone signaling).
Response: The specific functional categories of upregulated DEGs will be detailed in a separate manuscript of the Botrytis HZ-011’s transcriptome . Following weed-pathogen interactions, our focus in the weed transcriptome will be on the specific functional categories of downregulated DEGs.
- Interpret whether these upregulations represent fungal manipulation of host processes or residual host defense attempts.
Response: Same as above.
- Expand on how suppression of antenna proteins, carotenoid biosynthesis, and PSI/PSII components directly impacts energy metabolism and plant survival.
Response: This section has been expanded.
- Discuss how using Botrytis HZ-011 as a microbial herbicide compares with chemical herbicides in terms of sustainability, resistance risks, and safety.
Response: This section has been added.
- Acknowledge potential limitations.
Response: This section has been added.
Conclusion:
Please add a conclusion section.
Response: This section has been added.
We hope that our revised version will be satisfactory for publication in Microorganisms. Many thanks to reviewers and editor for the time and effort spent on this manuscript.
Sincerely yours,
Haixia Zhu
Reviewer 2 Report
Comments and Suggestions for Authors
The article presents interesting and well-designed transcriptomic research on the response of Chenopodium album to infection by Botrytis strain HZ-011. The introduction provides sufficient theoretical background and up-to-date references, which clearly justify the need for this study and highlight its relevance in the context of biological weed control. The methods are described in sufficient detail to allow reproducibility, covering inoculum preparation, inoculation conditions, RNA analysis procedures, and qRT-PCR validation. The research design appears appropriate for the posed questions, and the techniques applied are modern and consistent with current standards in transcriptomic studies. The results are presented in an organized manner, supported by statistical analyses and visualizations (graphs, heatmaps, diagrams). The conclusions logically follow from the results obtained and are well connected to previous literature. All tables and figures are clear and properly described. The English is understandable and generally correct; however, in some places the text could be simplified and stylistically unified to improve fluency. A deep language revision is not necessary, but minor stylistic improvements would be beneficial.
Additional comments and questions for the authors:
- Although the introduction contains a solid literature review, it would benefit from a stronger emphasis on the practical significance of the study – for example, what are the limitations of current bioherbicides, and how could Botrytis HZ-011 potentially overcome them?
- Some of the cited works are relatively old (e.g., 2002, 2003, 2005). The authors could supplement the review with more recent studies (2021–2024) that discuss transcriptomics in the context of weed control.
- Please clarify why the time points of 1, 4, and 5 days post-inoculation were chosen – are there preliminary data indicating that these are the most critical stages of infection?
- More details on the number of biological and technical replicates used in qRT-PCR would improve transparency.
- It is not entirely clear whether corrections for multiple testing (e.g., Benjamini–Hochberg method) were applied to the data, although FDR is mentioned – please specify how exactly this was done.
- In the bioinformatics section, it would be helpful to indicate the database versions used (KEGG, Nr, SwissProt) to ensure reproducibility.
- In some places, the results are extensive but not very synthetic. A summary table highlighting the main metabolic pathways and the corresponding number of up- and downregulated genes would be useful.
- The figures are clear, but in some cases (e.g., heatmaps) the labels are very small and hard to read. Increasing font size and using higher contrast colors would improve readability.
- In the Venn diagrams and trend analysis figures, the key genes identified as critical for herbicidal activity could be more clearly marked.
- The discussion could elaborate more on the differences between the chosen time points – which transcriptomic changes represent the initial defense response, and which reflect the later collapse of photosynthetic mechanisms?
- The discussion focuses primarily on photosynthesis – it would be valuable to also expand on other inhibited metabolic processes (e.g., amino acid metabolism, circadian rhythm, terpenoid biosynthesis).
- It would be helpful to reflect on whether the identified genes (PSB28, PSBP, CAP10A, CRTL-E-1) are unique to weeds such as C. album, or whether their suppression could pose a risk of unintended effects on crop species.
- The paper addresses transcriptome data, but does not discuss possible correlations with protein or metabolite levels. It would strengthen the study to acknowledge this as a limitation and point towards future proteomic or metabolomic validation.
- Please consider the potential influence of environmental conditions (e.g., abiotic stresses) on C. album’s response – would the same mechanisms operate under field conditions?
- The authors could more explicitly indicate how the discovered mechanisms might be practically applied in developing bioherbicides – for example, by using these genes as markers to evaluate pathogen efficacy in field trials.
- In some places the text is overly technical and contains very long sentences. Simplifying sentence structure and breaking down long paragraphs would improve readability.
- Avoid repetitions (e.g., the description of DEG numbers is repeated several times in a similar format). This could be shortened, with more focus placed on interpretation.
Author Response
Dear review 2,
Thank you very much for your comments and valuable suggestions on our manuscript, we have carefully made major revisions according to your comments. All changes have been clearly highlighted in the submited manuscript.
Below is our point by point response to these comments.
The article presents interesting and well-designed transcriptomic research on the response of Chenopodium album to infection by Botrytis strain HZ-011. The introduction provides sufficient theoretical background and up-to-date references, which clearly justify the need for this study and highlight its relevance in the context of biological weed control. The methods are described in sufficient detail to allow reproducibility, covering inoculum preparation, inoculation conditions, RNA analysis procedures, and qRT-PCR validation. The research design appears appropriate for the posed questions, and the techniques applied are modern and consistent with current standards in transcriptomic studies. The results are presented in an organized manner, supported by statistical analyses and visualizations (graphs, heatmaps, diagrams). The conclusions logically follow from the results obtained and are well connected to previous literature. All tables and figures are clear and properly described. The English is understandable and generally correct; however, in some places the text could be simplified and stylistically unified to improve fluency. A deep language revision is not necessary, but minor stylistic improvements would be beneficial.
Response: We sincerely thank the reviewers for dedicating their valuable time and effort to scrutinize our manuscript and for providing a series of highly professional, insightful, and constructive comments. All these suggestions have significantly enhanced the quality of our manuscript. We have thoroughly revised the manuscript according to your recommendations and respond to each point as follows:
Additional comments and questions for the authors:
- Although the introduction contains a solid literature review, it would benefit from a stronger emphasis on the practical significance of the study – for example, what are the limitations of current bioherbicides, and how could Botrytis HZ-011 potentially overcome them?
Response: Biological herbicides have established a solid foundation for developing environmentally friendly weed management solutions, earning them favor among researchers. This study aims to thoroughly investigate the application potential of Botrytis HZ-011 as a novel biological herbicide. The limitations of biological herbicides—such as environmental stability and rate of action—will be explored in subsequent research.
- Some of the cited works are relatively old (e.g., 2002, 2003, 2005). The authors could supplement the review with more recent studies (2021–2024) that discuss transcriptomics in the context of weed control.
Response: This section has been supplemented.
- Please clarify why the time points of 1, 4, and 5 days post-inoculation were chosen – are there preliminary data indicating that these are the most critical stages of infection?
Response: The selection of the 1, 4, and 5 days post-inoculation was based on preliminary results from microscopic observations and symptom records of the Botrytis HZ-011 infection process during preliminary experiments. Day 1 represents the initial onset of disease, while days 4–5 mark the peak symptom development phase. This design enables comprehensive capture of the entire pathological progression.
- More details on the number of biological and technical replicates used in qRT-PCR would improve transparency.
Response: This section has been supplemented.
- It is not entirely clear whether corrections for multiple testing (e.g., Benjamini–Hochberg method) were applied to the data, although FDR is mentioned – please specify how exactly this was done.
Response: Differential expression analysis was performed using the edgeR package. Raw p-values from the tests were corrected using the Benjamini-Hochberg method to control the false discovery rate (FDR). The threshold for significant differential expression was set at an FDR-adjusted p-value < 0.05 and |log2FoldChange| ≥1.
- In the bioinformatics section, it would be helpful to indicate the database versions used (KEGG, Nr, SwissProt) to ensure reproducibility.
Response: We sincerely appreciate your important suggestion. We fully agree that specifying the versions of bioinformatics databases used is crucial for the reproducibility of research. Following your recommendation, we have supplemented the bioinformatics section of the manuscript with the specific version information for all relevant databases, as detailed below: Nr database version: 20200514 KEGG data version: release101 Swiss-Prot database: 20200514
- In some places, the results are extensive but not very synthetic. A summary table highlighting the main metabolic pathways and the corresponding number of up- and downregulated genes would be useful.
Response: Analysis of metabolic pathways and the corresponding number of differentially expressed genes has been added to the manuscript.
- The figures are clear, but in some cases (e.g., heatmaps) the labels are very small and hard to read. Increasing font size and using higher contrast colors would improve readability.
Response: This section has been modified.
- In the Venn diagrams and trend analysis figures, the key genes identified as critical for herbicidal activity could be more clearly marked.
Response:We appreciate the valuable suggestions provided by the reviewers. Regarding labeling key genes in the figure, after careful consideration, we believe that directly labeling a large number of gene names within the figure would result in extreme crowding and illegibility, thereby compromising the figure's visual clarity and the effective communication of information. We will provide additional details on this aspect within the manuscript.
- The discussion could elaborate more on the differences between the chosen time points – which transcriptomic changes represent the initial defense response, and which reflect the later collapse of photosynthetic mechanisms?
Response: This section has been supplemented.
- The discussion focuses primarily on photosynthesis – it would be valuable to also expand on other inhibited metabolic processes (e.g., amino acid metabolism, circadian rhythm, terpenoid biosynthesis).
Response: We are grateful for the valuable suggestions provided by the reviewers. In this study, we will focus our discussion on the most direct and rapid initial target—the photosynthetic mechanism itself. However, a more in-depth analysis of these secondary metabolic pathways will undoubtedly be an important direction for future research to comprehensively elucidate the multidimensional physiological collapse induced by HZ-011.
- It would be helpful to reflect on whether the identified genes (PSB28, PSBP, CAP10A, CRTL-E-1) are unique to weeds such as C. album, or whether their suppression could pose a risk of unintended effects on crop species.
Response: The issue you raised regarding potential non-target effects is a critical step in the transition of any herbicide from laboratory development to practical application. We fully agree with your perspective and have prioritized conducting tests on crops as the most crucial next step in this research.
- The paper addresses transcriptome data, but does not discuss possible correlations with protein or metabolite levels. It would strengthen the study to acknowledge this as a limitation and point towards future proteomic or metabolomic validation.
Response: Agree. This section has been supplemented.
- Please consider the potential influence of environmental conditions (e.g., abiotic stresses) on C. album’s response – would the same mechanisms operate under field conditions?
Response: We appreciate the reviewer's insightful comment. Indeed, multiple abiotic stresses in field environments—such as drought, high temperatures, and nutrient deficiencies—may significantly impact the physiological state of C. album and its interaction outcomes with Botrytis HZ-011. This represents a critical consideration when evaluating its practical application potential.
- The authors could more explicitly indicate how the discovered mechanisms might be practically applied in developing bioherbicides – for example, by using these genes as markers to evaluate pathogen efficacy in field trials.
Response: Agree. This section has been supplemented. The molecular mechanisms elucidated in this study not only reveal the pathogenicity of Botrytis HZ-011 but also provide direct molecular tools and insights for its development and optimization in practical agricultural applications.
- In some places the text is overly technical and contains very long sentences. Simplifying sentence structure and breaking down long paragraphs would improve readability.
Response: This section has been modified.
- Avoid repetitions (e.g., the description of DEG numbers is repeated several times in a similar format). This could be shortened, with more focus placed on interpretation.
Response: This section has been modified.
We hope that our revised manuscript will be reviewed and commented on by peer experts. Many thanks to reviewers and editor for the time and effort spent on this manuscript.
Sincerely yours,
Haixia Zhu
Reviewer 3 Report
Comments and Suggestions for Authors
The manuscript contains quite interesting results on the use of pathogens to control weeds, in this case Chenopodium album, which is a serious threat to the cultivation of many plant species, including sugar beet, potato, and soybean. What is the effectiveness of the tested solution? When do the Authors believe their research results can be used to produce microbial herbicides? What problems still need to be resolved? Which research topics do the authors intend to address in the near future?
Materials and Methods
This chapter is very carefully prepared. Please provide the location and duration of the research.
Results
These are well-described. The tables and figures are clear and understandable.
Discussion
It is very short and needs to be expanded.
Conclusions
None
References
The number of publications is quite small (only 27). Some are over 10 years old. I suggest deleting these and supplementing them with more recent publications.
Author Response
Dear review 3,
Thank you very much for your comments and valuable suggestions on our manuscript, we have carefully made major revisions according to your comments. All changes have been clearly highlighted in the submited manuscript.
Below is our point by point response to these comments.
The manuscript contains quite interesting results on the use of pathogens to control weeds, in this case Chenopodium album, which is a serious threat to the cultivation of many plant species, including sugar beet, potato, and soybean. What is the effectiveness of the tested solution? When do the Authors believe their research results can be used to produce microbial herbicides? What problems still need to be resolved? Which research topics do the authors intend to address in the near future?
Response:Thank you for pointing this out. We will test efficacy through host range determination and field plot trials. The pathogen will be inoculated onto non-target crops such as sugar beet, potato, and soybean to confirm its specific infection only to Chenopodium, while ensuring safety for agricultural crops. Small plots will be established in natural field environments to evaluate the pathogen's weed control efficacy, persistence, and actual impact on crop yields under real-world conditions (e.g., varying temperatures, humidity, and light exposure). Once field trial results demonstrate stable and reproducible outcomes, the findings can be applied to produce microbial herbicides. Remaining challenges include: large-scale production and formulation, stability testing under environmental factors, safety reassessment, and integration with existing agricultural practices. Our future research will focus on optimizing fermentation and formulation processes, conducting in-depth field efficacy studies, investigating mode of action, and examining environmental fate and ecological impacts.
- Materials and Methods
This chapter is very carefully prepared. Please provide the location and duration of the research.
Response: This section has been supplemented.
- Results
These are well-described. The tables and figures are clear and understandable.
Response:Thank you for your affirmation and encouragement.
- Discussion
It is very short and needs to be expanded.
Response: This section has been supplemented.
- Conclusions
None
Response: This section has been supplemented.
- References
The number of publications is quite small (only 27). Some are over 10 years old. I suggest deleting these and supplementing them with more recent publications.
Response: This section has been supplemented as required.
We hope that our revised version will be satisfactory for publication in Microorganisms. Many thanks to reviewers and editor for the time and effort spent on this manuscript.
Sincerely yours,
Haixia Zhu
Round 2
Reviewer 1 Report
Comments and Suggestions for Authors
Dear Authors,
Thank you for making significant improvements to the manuscript. I would like to suggest the following minor revisions:
- Please provide a reference for 25 mL per pot.
- Were genes selected randomly, or based on specific pathways?
Author Response
Dear review,
Thank you for your suggestion, we have carefully made minor revisions according to your comments.
Please provide a reference for 25 mL per pot.
Response: Based on our experience, spraying 25 mL per pot is sufficient to thoroughly moisten the entire plant until droplets form on the stems and leaves.
Were genes selected randomly, or based on specific pathways?
Response: Thank you for pointing this out. Gene selection was not random but based on systematic screening results derived from transcriptome data analysis and KEGG pathway enrichment analysis. Genes significantly downregulated in photosynthesis-related metabolic pathways, PSB28, PSBP, CAP10A, and CRTL-E-1 exhibited highly significant expression changes post-treatment. Consequently, these were designated as candidate genes for subsequent functional validation.
Thanks again for your time spent on this manuscript.
Haixia